# Are There Thresholds in Glioblastoma Cell Death Responses Triggered by Temozolomide?

**DOI:** 10.3390/ijms20071562

**Published:** 2019-03-28

**Authors:** Yang He, Bernd Kaina

**Affiliations:** Institute of Toxicology, University Medical Center, Obere Zahlbacher Str. 67, D-55131 Mainz, Germany; yanghe01@uni-mainz.de

**Keywords:** DNA damage, O6-alkylguanine, temozolomide, dose-response curves, time-response, glioblastoma, glioma, apoptosis, autophagy, senescence, p53

## Abstract

Temozolomide (TMZ) is an alkylating agent used in the treatment of high-grade malignant glioma, notably glioblastoma multiforme, the most aggressive form of brain cancer. The drug induces a dozen DNA methylation adducts, including *O*^6^-methylguanine (*O*^6^MeG), which is the most toxic primary DNA lesion as it causes the formation of DNA double-strand breaks (DSBs) that trigger apoptosis. In p53 wild-type cells, TMZ activates p-p53ser15 and p-p53ser46, which have opposing dual functions regulating survival and death, respectively. Since the use of TMZ in a therapeutic setting is limited because of its side effects, the question arises as to the existence of threshold doses that activate the death pathway and start apoptosis. To determine whether there is a threshold for the TMZ-induced DNA damage response and exploring the factors regulating the switch between p53 dependent survival and death, the glioblastoma lines LN-229 (deficient in MGMT) and LN-229MGMT (stably transfected with MGMT) were exposed to different doses of TMZ. p53 protein expression and phosphorylation levels of p-p53ser15 and p-p53ser46 were determined by Western blotting. Also, apoptosis, senescence and autophagy levels were checked after different doses of TMZ. The results show that pro-survival p-p53ser15 and pro-death p-p53ser46 were induced by *O*^6^MeG in a specific dose- and time-dependent manner. p-p53ser15 was an early response while p-p53ser46 was activated at later times following treatment. Unexpectedly, the dose-response curves for total p53, p-p53ser15 and p-p53ser46 were linear, without an obvious threshold. *O*^6^MeG induces apoptosis late after treatment as a linear function of TMZ dose. This was observed for both p53 proficient LN-229 and p53 lacking LN-308 cells. A linear dose-response after TMZ was also observed for senescence and autophagy as well as γH2AX, an indicator of DSBs that are considered to be the downstream trigger of apoptosis, senescence and autophagy. LN-229MGMT cells were highly resistant to all measured endpoints because of repair of the critical primary lesion. Although LN-308 were less responsive than LN-229 to TMZ, they displayed the same TMZ-induced DSB level. The observed linear dose-responses are not compatible with the view that low DNA damage level evokes survival while high damage level activates death functions. The data bear important therapeutic implications as they indicate that even low doses of TMZ may elicit a cytotoxic response. However, since *O*^6^MeG triggers apoptosis, senescence and autophagy in the same dose range, it is likely that the accumulation of senescent cells in the population counteracts the killing effect of the anticancer drug.

## 1. Introduction

In cancer therapy, chemotherapeutics with genotoxic activity are being used routinely. It is well known that these anticancer drugs induce DNA damage that triggers complex cellular DNA damage responses (DDR), which determine the fate of the cell, making the decision between survival and death [1]. Key players involved in this scenario are the DDR kinases ATM, ATR and DNA-PK, the checkpoint kinases CHK1, CHK2, the stress kinase HIPK2, and further downstream, the transcription factor and tumor suppressor protein p53. During activation, p53 becomes phosphorylated, liberates from its inhibitor MDM1, becomes stabilized and binds as a transcription factor in a dimeric form to a p53 consensus sequence in the promoter of pro- and anti-apoptotic genes [2]. In addition, p53 has other functions that are independent of transcriptional activation of genes [3]. It is generally believed that low DNA damage levels activate pro-survival and high damage levels activate pro-death genes and cellular functions [4,5,6]. For example, p53 stimulates the transcription of p21, which results in cell cycle arrest, and DNA repair genes such as DDB2, which enhances the repair capacity, leading to removal of toxic DNA lesions from DNA [7]. At high dose levels p53 turns into a “killer” through activation of pro-death functions such as the proapoptotic genes Bax, Bak and Fas [2]. Although this concept derived mostly from work with ionizing radiation is reasonable, there is not much experimental proof of it for chemical genotoxins, notably anticancer drugs. The concept implicates that there are threshold doses for cell death, i.e., low doses do not elicit activation of apoptosis pathways while high doses do.

A proof of this concept requires maximum understanding of the cell death pathways activated by a given genotoxicant. A well-studied drug in this respect is temozolomide (TMZ), which is used in first-line therapy for high-grade gliomas, including astrocytoma (WHO ^0^3) and glioblastoma multiforme (glioma WHO ^0^4) [8]. The main target of TMZ is the nuclear DNA in which, similar to other S_N_1 alkylating agents, at least 12 nucleophilic sites can become methylated [9]. The major methylation products are *N*-methylpurines such as *N7*-methylguanine, *N3*-methylguanine and *N3*-methyladenine, while *O*-methylpurines are less frequent. Thus, *O*^6^-methylguanine (*O*^6^MeG) accounts for maximally 7% of the total methylations [9]. Although produced in minor amounts, the damage is highly genotoxic and cytotoxic if not repaired by the suicide enzyme *O*^6^-methylguanine-DNA methyltransferase (MGMT) [10]. If cells are repair competent, *O*^6^MeG is quickly removed from DNA. Under this condition, cells become highly resistant to *O*^6^-alkylating agents and higher doses of a methylating agent are required to achieve a killing effect, which results from saturation of base excision repair and repair by ALKB homologous proteins (ALKBH) [11]. Therefore, in the high dose setting, other lesions than *O*^6^MeG, which are less toxic, give rise to cell death. The doses of TMZ in a therapeutic setting are very likely too low to achieve cell death resulting from non-repaired N-alkylations. Therefore, with an achievable serum concentration of up to 50 µM TMZ, the *O*^6^MeG response plays a key role in determining tumor cell death.

The mechanisms of *O*^6^MeG triggered genotoxic responses have been described previously [11]. In brief, *O*^6^MeG is a mutagenic mispairing lesion that results in mismatches with thymine that are subject to mismatch repair (MMR). Reinsertion of thymine during MMR causes a futile MMR cycle with gapped DNA that finally gives rise to DNA replication blockage and the formation of replication-mediated DNA double-strand breaks (DSBs), which occurs in the post-treatment cell cycle [12]. These events provoke the activation of ATR und ATM, and downstream CHK1 and CHK2, respectively, as well as p53 phosphorylation [13].

Upon genotoxic stress, p53 can be phosphorylated at different sites. p53 phosphorylated at serine 15 (p53ser15) and serine 20 (p53ser20) results from ATM/ATR-CHK2/CHK1 activation, while phosphorylation at serine 46 (p53ser46) results from activation of the kinase HIPK2 (for review, see [14]). We have recently shown that this also occurs in glioblastoma cells upon treatment with TMZ. We also showed that p53ser46 exerts a pro-apoptotic function as downregulation of HIPK2, the kinase responsible for this phosphorylation, attenuated significantly the level of apoptosis in TMZ-treated LN-229 glioblastoma cells [15].

In light of the hypothesis outlined above, according to which low doses elicit pro-survival and high doses pro-death functions, we wondered whether the dose-response of key players of DDR shows the hypothesized threshold. Here, we present data showing the non-existence of threshold doses for γH2AX, p53ser15, p53ser46, apoptosis, autophagy and senescence in the p53 expressing LN-229 glioblastoma cell system.

## 2. Results

### 2.1. Dose Response and Time Dependence of p53, p-p53Ser15 and p-53Ser46 in Temozolomide Treated Glioblastoma Cells

To determine the dose-response of total p53, the pro-survival form p-p53Ser15 and the pro-death form p-53Ser46, Western blot experiments were performed using the line LN-229. This glioblastoma line bears p53 that harbours a point mutation, which however retains the trans-activating activity of p53 [16]. Therefore, the line can be considered functionally p53 wild-type. LN-229 is MGMT deficient [17]. We further used the isogenic line LN-229MGMT, which was stably transfected with human MGMT cDNA and thus expresses MGMT [18], in order to find out whether the responses observed are triggered by the specific TMZ-induced DNA damage *O*^6^MeG. The p53 expression levels were determined 24 and 72 h after the addition of TMZ to the medium of exponentially growing cells. As shown in Figure 1A, the total p53 and p-p53ser15 protein levels went up with increasing TMZ doses 24 h after treatment while the p-p53ser46 protein level remained nearly unaffected. With the highest dose of TMZ in this experimental series (125 μM) the total p53 level was 3.7 times above the control group while the expression level of p-p53ser15 reached 10.9 times above the control level where p-p53Ser15 was only marginally detected (Figure 1A for a representative blot and for quantification Figure 1C). The finding that, in contrast to p-53Ser15, p-p53ser46 was not induced 24 h after treatment indicates that there is a sequential activation of pro-survival and pro-death factors in glioblastoma cells exposed to TMZ. In LN-229MGMT, we did not observe any significant increase in p53, p-p53ser15 and p-p53ser46 (Figure 1A, right panel), demonstrating that the effects observed were triggered by the *O*^6^MeG lesion, which is specifically repaired by MGMT.

After 72 h TMZ treatment, LN-229 cells displayed increasing signals of p53 and p-p53ser46 (Figure 1B), p-p53ser15 also increased above the control, although this was moderate compared to p53 and p-p53ser46 (Figure 1B and for quantification Figure 1D). With the highest dose tested, total p53 accumulated 13.1 times over the control, p-p53ser46 reached 8.9 times, and p-p53ser15 was 5.2 times the control level. In LN-229MGMT cells, p-53Ser46 was not enhanced and p-p53ser15 was only slightly activated, very likely due to residual amounts of non-repaired *O*^6^MeG (Figure 1B, right panel). Overall, the data revealed that p-p53Ser15 is an early response and p-53Ser46 a late response triggered by the TMZ-induced lesion *O*^6^MeG.

### 2.2. Dose Response and Time-Course of Apoptosis Induction

To compare the sequence of anti- and pro-apoptotic p53 with cell death induction, we measured apoptosis and necrosis in LN-229 and LN-229MGMT after TMZ treatment in a dose-range up to 125 µM by AV/PI staining and flow cytometry. Although 72 h after exposure to TMZ, the p-p53ser46 level was significantly enhanced (Figure 1B), the apoptosis level stayed low and was not significantly above the control level (Figure 2A). Apoptosis in LN-229 started to increase 96 h after TMZ exposure, and reached its maximum at 120 and 144 h (Figure 2A,E,F). This confirms previous observation that *O*^6^MeG triggered apoptosis is a late response. Of note, the cells were kept in an exponentially growing state throughout the whole examination period. Apoptosis in LN-229 reached a saturation level at >50 µM, presumably because of other endpoints that were concomitantly induced such as senescence and autophagy [19]. Necrosis (defined as AV+PI+) was only marginally induced (Figure 2C), supporting our previous finding that *O*^6^MeG is a powerful apoptotic lesion. In LN-229MGMT cells, neither apoptosis nor necrosis was induced with all doses and time points assayed (Figure 2B,D), demonstrating that *O*^6^MeG was responsible for the cytotoxic effects observed.

### 2.3. Is There a Threshold in p53, p-p53ser15 and p-p53ser46 Induction?

Next, we addressed the question of whether there is a threshold in the dose-response in LN-229 cells. For this experimental series, we applied TMZ in a low dose range of up to 20 µM. Cells were harvested from exponentially grown cultures 24 h and 72 h after the onset of treatment and total cell extracts were subjected by Western blot analysis. As shown in Figure 3A, in the low dose range and measured 24 h after TMZ exposure, p-p53ser15 was the most sensitive indicator for DNA damage, which increased with increasing dose. After 72 h exposure, the p-p53ser46 level was also increasing (Figure 3B). As expected on the basis of previous results, in LN-229MGMT all effects were vanished (Figure 3A,B left panels). The results indicate that after low dose TMZ treatment, the pro-survival factor p-p53ser15 is phosphorylated first (and quite early) compared to the pro-death factor p-p53ser46, which gets activated at a later stage.

Originally, we suspected that the dose-response for the pro-apoptotic p-p53ser46 would show a threshold. This, however, was not the case. As revealed by the quantification in Figure 3C,D, there is a linear increase in the amount of p53, p-p53Ser15 and p-p53Ser46. The total p53 level already reached saturation with a dose of 5 µM. The increase of p-p53ser15 (24 h) and p-p53ser46 (72 h) was linear over the whole dose range tested.

It is also interesting that after 72 h, the p-p53Ser15 dropped to the control level (Figure 3D), indicating this is an early and transient response compared to p-53Ser46, which is a late (Figure 3D, see also Figure 1) and presumably also long-lasting response.

### 2.4. Is There a Threshold in Apoptosis Induction?

Having shown that p-p53Ser46 increases linearly with dose, we measured the dose-response of apoptosis (and necrosis) in LN-229 cells in the same low dose range (0–20 µM TMZ). As shown in Figure 4A, there is a linear increase (best fit) in the level of apoptosis without any obvious threshold dose. The dose that displayed a significant increase above the control level was 2.5 µM. Again, necrosis was not significantly induced (not shown) and MGMT expressing cells were effect-negative (Figure 4B).

To explore the possibility that p53 is responsible for the lack of a no-effect threshold, another glioma cell line, LN-308, was introduced in this step of analysis. LN-308 is completely lacking p53 (Figure 4C) due to gene deletion [16]. It is also MGMT deficient (Appendix A). Nevertheless, in order to avoid any effects caused by residual MGMT not detectable by the assays, we routinely pre-treated the cells with *O*^6^BG. The data shown in Figure 4D revealed that LN-308 cells are more resistant than LN-229 to TMZ-induced apoptosis. The best fit of the dose-response curve was linear and did not reveal a threshold.

To verify the data, colony formation assays were employed, which are considered to be highly sensitive for measuring reproductive cell death. After exposing LN-229, LN-229MGMT and LN-308 cells plated on dishes to TMZ and allowing them to grow for about two weeks, the formed colonies were counted. The survival fraction declined dose-dependently in LN-229 and LN-308. For LN-229 cells, the survival curves did not show a shoulder (Figure 5), supporting the view that there is no threshold for the induction of cell death in this line. In the dose range used, LN-229MGMT cells were strongly protected against TMZ-induced cell death, showing only a slight decline in their colony-forming ability (Figure 5). For LN-308 cells, the survival curve displayed only a marginal shoulder (Figure 5 and Appendix A). Overall, the data confirmed the results obtained in the apoptosis assays.

### 2.5. Dose-Response of TMZ-Induced Senescence and Autophagy

In a previous work, we showed that TMZ induces not only apoptosis, but also senescence and autophagy in LN-229 cells [19]. Therefore, we wondered whether senescence and autophagy are induced in the same dose range as apoptosis and whether there is a threshold in the dose-response curves. For comparison, senescence and autophagy were also determined in p53 deficient LN-308 cells. Senescence induced by TMZ increased as a linear function of dose (Figure 6A). Similar to apoptosis, MGMT protected against TMZ-induced senescence (Figure 6B), indicating the primary trigger is *O*^6^MeG. In LN-308 cells, the slope of the senescence curve was lower than in LN-229 cells, and therefore, comparatively high doses of TMZ have to be used to significantly induce senescence. Extrapolation of the dose-response curves indicated the non-existence of a threshold (Figure 6C).

The autophagy response was similar to senescence. TMZ induced more autophagy in LN-229 than in LN-308 cells (Figure 7A,C), while LN-229MGMT showed no response at all (Figure 7B). Taken together, the results show that p53 proficient LN-229 cells are more sensitive than p53 deficient LN-308 cells as to TMZ-induced apoptosis, senescence and autophagy. No threshold was observed for these endpoints in LN-229 and LN-308 cells.

### 2.6. Dose-Response of TMZ-Induced DNA Double-Strand Breaks

The endpoints we have assessed are triggered by the primary lesion *O*^6^MeG, whose processing gives rise to DSBs that are considered to be the ultimate trigger of p53 stabilization and cell death [13]. Therefore, different cell death responses could be the result of differences in DSB formation. Thus, the more sensitive p53 proficient LN-229 cells might exhibit better *O*^6^MeG processing resulting in DSBs than LN-308 cells upon treatment with TMZ. To explore this hypothesis, the γH2AX foci assay was used, which is an accepted indicator of DSBs [20,21]. The cells (LN-229, LN-229MGMT and LN-308) were treated with 0-20 μM TMZ and, 72 h later, the γH2AX foci were determined and quantified. The γH2AX foci in LN-229 showed a liner increase with dose of TMZ (Figure 8A) and no threshold was obvious. In LN-229MGMT cells, DSBs were not induced at significant level in the whole dose range (Figure 8B). In LN-308, the DSB frequency was very similar to LN-229. The increase was linear and, again, no threshold could be detected (Figure 8C). The similar frequencies of DSBs in LN-229 and LN-308 indicate that sensitivity differences between the cell lines are the result of downstream events triggered by DSBs, presumably through p53.

## 3. Discussion

The DNA methylating agent temozolomide is a first-line drug in the treatment of high-grade malignant glioma. It is effective in inducing cell death if the tumor lacks MGMT or expresses it at low level, i.e., < 30 fmol/mg protein [22]. These tumors are defined as “methylated” because of MGMT promoter CpG methylation, which correlates with silencing of the gene [23] and deficient or low MGMT protein expression and enzyme activity [24]. Since *O*^6^MeG induced by TMZ (and other methylating anticancer drugs) is a toxic DNA damage, it is understandable that MGMT deficiency (determined, e.g., by promoter methylation) leads to responsiveness of the tumor [25,26]. Despite these well-known relationships, the prognosis of glioblastoma, which account for up to 70% of high-grade malignant glioma, is bleak as the median length of survival is only 14.6 months (12.6 and 23.4 months in the MGMT-unmethylated and MGMT-methylated subgroups, respectively) [27]. Although recent phase III clinical trials showed that the median overall survival for adult patients with newly diagnosed glioblastoma can reach up to 20 months in the control cohorts, indicating a trending increase in median overall survival, the prognosis is still bad with 5-year overall survival rates of less than 10% (for references see [28]). Treatment with TMZ occurs daily along different schedules [29,30,31,32]. The serum concentration of TMZ has been determined to be in the range of up to 30 µM, with a half-life of about 2 h [33,34,35,36,37,38]. In a therapeutic setting with a single oral dose of 150 mg/m^2^, the peak plasma concentration was, on average, 28.4 µM (5.5 µg/mL) and the brain interstitium concentration 1.5 µM (0.3 µg/mL) [39]. In another study TMZ was determined following oral 200 mg/m^2^ TMZ, with a plasma peak level of 72 µM and a cerebrospinal fluid level of 9.9 µM [40]. Thus, the TMZ concentration at the target organ seems to be rather low and it is reasonable to suppose, notably in view of the high recurrence rate, that the TMZ level is not high enough in order to exert a killing effect on residual (post-operative) glioblastoma cells. This notion is fueled by the supposition that at low dose levels cell death is not induced, which goes back to the general paradigm that low DNA damage levels induce survival functions, whereas high DNA damage levels activate cellular death pathways [1,2,3,4,5,6]. This view implies that DNA damage thresholds do exist that regulate the balance between life and death. This work was aimed at proving or disproving this widely accepted hypothesis.

First, we have shown that in LN-229 and LN-308 glioblastoma cells, which are functionally wild-type and mutant for p53, respectively [16], the amount of DSB (γH2AX foci) increases as a linear function of dose. TMZ does not need metabolic activation. It spontaneously decomposes, yielding carbenium ions that methylate DNA dose-dependently. From this it is reasonable to conclude that *O*^6^MeG is induced as a linear function of dose. The linear dose-response for DSBs indicates that the rate of conversion of *O*^6^MeG into DSB is independent on dose, and there is no defense at low dose levels that prevents the formation of DSBs in LN-229 and LN-308 cells. In this model system, we determined about 60 DSBs with a dose of 20 µM TMZ. The amount of *O*^6^MeG induced under these conditions is not known.

We further show in Western blot experiments that p53, p-p53ser15 and p-p53ser46 increase up to a saturation level as a linear function of dose. This was a surprising finding since it collides with the view that low DNA damage triggers survival and high DNA damage triggers death functions. Upon TMZ treatment, p-p53ser15 results from ATR (ATM) and downstream CHK1 (CHK) activation [41], which is likely the result of blocked replication forks and DSBs formed on collapsed forks. The linear dose-response suggests that even low *O*^6^MeG and DSB levels induced by TMZ are able to activate the DNA damage checkpoint kinases that phosphorylate 53 at serine 15. Unexpectedly, p-p53ser46 was also generated at low dose levels without a detectable threshold. p-p53ser46 results from activation of the stress kinase HIPK2 [42]. The pathway of HIPK2 activation in general [43] and following TMZ in glioblastoma cells, including LN-229, has been described [15]. The available data suggest that primarily ATR and following secondary activation, also ATM, phosphorylate SIAH1, the inhibitor of HIPK2 [44,45]. This leads to degradation of SIAH1 and liberation and stabilization of HIPK2, which in turn phosphorylates p53 at serine 46 (see Figure 9). The linearity of p-p53ser46 accumulation indicates that ATR (ATM) is able to phosphorylate SIAH1 and thus liberate HIPK2 even at very low TMZ doses.

We have shown that p-p53ser15 and p-p53ser46 become activated following TMZ treatment in a sequential order, with early activation of p-p53ser15 and late activation of p-p53ser46. p-p53ser15 becomes detectable 24 h after treatment and declines a day later, indicating that this is a transient response. p-p53ser46 was detected 3 days after treatment and was still detectable when cells started to undergo apoptosis. This is in line with the pro-apoptotic role of this phosphorylated form of p53.

In accordance with this is the finding that apoptosis (Figure 4) and reproductive death (Figure 5) of LN-229 cells do not display a clear no-effect threshold. Cell death appears to be a linear function of the dose of TMZ. It is known that p-p53ser46 binds to the promoter of pro-apoptotic genes, including the death receptor FAS (alias CD95/APO1), and thus stimulates its transcription [46]. This was recently shown to occur upon TMZ in LN-229 cells [15]. Obviously, there is no threshold for p-p53ser46 transactivation activity in this cell system.

Similar to apoptosis, DNA damage-induced senescence (Figure 6) and autophagy (Figure 7) were induced as a linear function of the dose of TMZ. In MGMT expressing cells, TMZ was ineffective in inducing these effects (doses up to 50 µM) suggesting that they were triggered by *O*^6^MeG. Previously, we have shown that senescence and autophagy are regulated by the same upstream damage response pathway that regulates apoptosis [19]. The linearity for these endpoints indicates that pro- and anti-death functions are induced simultaneously at each dose level. From the therapeutic point of view, the finding points to the need of inhibiting the pro-survival functions senescence and autophagy in a way that cells preferentially enter the death pathway.

Finally, we observed that the p53 deficient glioblastoma cell line, LN-308 is more resistant to the induction of apoptosis by TMZ. This finding is compatible with our previous observations according to which MGMT deficient p53wt glioma cells are more sensitive to the cytotoxic (apoptotic) effect of TMZ than MGMT deficient p53 mutated cells that lack the transactivation activity of p53 [47]. The p-53 independent apoptotic pathway of glioblastoma cells is bound on the endogenous mitochondrial pathway, which seems to be more refractory than the p53 regulated death receptor pathway [18]. Of note, p53 also stimulates the mitochondrial cell death route by supporting the translocation of BAX to the outer mitochondrial membrane and sequestering Bcl-2, leading to cytochrome C release and apoptosome formation [48,49]. It is therefore reasonable to conclude that p53-driven apoptosis rests on both p-p53ser46 promoter activation and exacerbation of mitochondrial damage through cytoplasmatic p53.

## 4. Conclusions and Implications

Although we are aware that this study needs extension to other cell lines and tumor models, LN-229 provides an example where the paradigm that low doses activate survival and high doses death functions does not apply. Regarding DNA repair, it is known that p-p53ser15 triggers the activation of DNA repair genes, which causes protection against genotoxins [7]. According to our experience with different cell types and genotoxins, the most robust p53-stimulated repair genes encode DDB2 and XPC as well as the translesion polymerase Pol eta (Pol H) [50]. However, these genotoxic stress-inducible repair proteins are not involved in the repair of TMZ-induced DNA methylation damage. A reasonable candidate for causing a threshold is MGMT. Thus, from work that included bacteria to humans, it became clear that MGMT mediated DNA repair gives rise to a mutagenic and toxic threshold [11,51,52]. However, a search for an adaptive response in brain cancer cells revealed that MGMT is not inducible by TMZ, which is clearly different from rodent cells in which MGMT was shown to be upregulated following genotoxic stress [53] in a p53 dependent manner [54]. Therefore, lack of induction of repair of *O*^6^MeG in glioma cells is surely a contributing factor for the non-existence of a threshold. If ATR/ATM becomes activated even with low damage levels and also triggers senescence, autophagy and apoptosis, the important question arises as to the mechanism that makes the switch between the pathways. This is clearly an attractive area of future research.

In view of the limited amount of cell lines used in this study, it is too early for clinical implications. Nevertheless, the data may be taken to indicate that even a low dose of TMZ is able to elicit a cytotoxic response in p53 wild-type and MGMT lacking tumors. Of note, a prerequisite for *O*^6^MeG induced cytotoxicity is cell proliferation. If a fraction of tumor cells is released in a senescent state, it will no longer be subjected to *O*^6^MeG triggered apoptosis. This might especially be the case if cells are treated repetitively. Therefore, on the basis of the results presented here, the metronomic dose protocol (drug application at low and frequent doses) bears beneficial effects by exacerbating cytotoxicity, but also adverse effects since the fraction of non-proliferating (senescent) cells might be increasing with each consecutive treatment dose. It should also be considered that TMZ is usually given concomitantly with ionizing radiation (usually 2 Gy per treatment), which may additionally ameliorate the fraction of non-proliferating tumor cells. If the arrest state is transient, it is conceivable that the fraction of senescent cells at the end of therapy contributes to recurrence, which is usually the unfortunate case for glioblastomas. Again, we are aware of the limitations of the study, which rests on comparison of only three cell lines (LN-229, LN-220MGMT and LN-308). It provides, however, an example of lack of threshold doses in cell death responses (γH2AX, p53ser15 and p53ser46, apoptosis, autophagy and senescence) if MGMT is lacking. The data warrant further studies with a larger set of well-defined cell lines, stem cells and tumors in situ prior to and after therapy.

## 5. Materials and Methods

### 5.1. Cell Lines and Culture Conditions

The human glioma cell line LN-229 was purchased from American Type Culture Collection (ATCC), the human glioma line LN-308 were a generous gift from Prof. Dr. M. Weller (Laboratory of Molecular Neuro-oncology, University of Zurich, Switzerland). Upon receipt, the cells were amplified for cryopreservation in liquid-N_2_ and freshly thawed cell stocks were used for every battery of tests. LN-229, LN-308 and the LN-229MGMT transfected cells [19] were cultured in DMEM (Gibco, Life Technologies Corporation, Paisley, UK) supplemented with 10% FBS and penicillin/streptomycin (PAA Laboratories, GmbH, Cölbe, Germany). Cells were maintained at 37°C in a humidified 5% CO_2_ atmosphere.

### 5.2. Cell Seeding and Growth

Cells were cultured in DMEM supplemented with 10% fetal bovine serum. Cells were seeded 24 h before any treatment to settle down and get ready for knockdown and treatments. Seeding density was such that exponential cell growth was ensured for the whole experimental period.

### 5.3. Drugs and Drug Treatment

The MGMT inhibitor *O*^6^-benzylguanine (*O*^6^BG, Sigma-Aldrich, Steinheim, Germany) was dissolved in DMSO to a stock concentration of 10 mM, aliquoted and stored at −20 °C. To inactivate any residual MGMT, 1 h before the addition of TMZ *O*^6^BG was added to the medium. The final concentration of *O*^6^BG in DMEM was 10 μM. Temozolomide was a generous gift of Dr Geoff Margison (University of Manchester, UK). Stocks were dissolved in dimethyl sulfoxide (DMSO, Carl Roth GmbH, Karlsruhe, Germany), diluted in two parts sterile dH_2_O to a concentration of 35 mM, aliquoted and stored at −80 °C until use. After thawing, the stock solution was sonicated for 10 s to help TMZ dissolution. Cells were exposed to TMZ by directly adding the aqueous TMZ stock solution to the medium.

### 5.4. Colony Survival Assays

Cells were seeded in 6 cm dishes, treated 1 day later with TMZ and left to grow in a CO_2_ incubator until colonies appeared (microscopic control). Colonies were fixed in methanol and stained (1.25% Giemsa, 0.125% crystal violet). The plating efficiency represents the number of colonies formed in the control sample/ number of cells seeded in the control sample × 100%, and the surviving fraction is the number of colonies after treatment/number of cells seeded x PE. Colonies containing more than about 50 cells were scored.

### 5.5. Apoptosis/Necrosis Flow Cytometry

For the determination of apoptosis and necrosis the annexin V/propidium iodide (AV/PI) assay coupled with flow cytometry analysis was used. In brief, for harvest cells in the supernatant were collected in a 15 mL tube, samples were washed twice with PBS and detached with trypsin/EDTA solution. They were washed twice in PBS and 50 μL 1× binding buffer and 2.5 μL Annexin V/FITC (Miltenyi Biotec GmbH, Bergisch Gladbach, Germany) were added to each sample. Following 15 min incubation in the dark on ice, 430 μL 1x binding buffer and 1 µg/mL PI (Sigma-Aldrich, Steinheim, Germany) were added to the cells. Data acquisition was done by a FACS Canto II flow cytometer (Becton Dickinson GmbH, Heidelberg, Germany). Annexin V positive cells were classified as apoptotic while annexin V and PI double-positive cells were classified as necrotic/late-apoptotic. The data were analysed using the BD FACSDiva software. A representative plot of control and treated cells is shown in Appendix A.

### 5.6. Whole-Cell Protein Extracts

Cells were washed twice with PBS and 300-600ul RIPA buffer was added to each sample. The cells were scraped off and transferred to pre-cooled tubes, vortexed and put on ice. Sonication was employed for disrupting cells (3 × 10 pulses) and samples were centrifuged (10 min at 4 °C, 14,000 rpm) to obtain the protein extract in the supernatant. Protein concentration was determined by Bradford. The extraction buffer and RIPA buffer recipes were as follows: Extraction buffer: 20 mM tris(hydroxymethyl)aminomethane [TRIS] HCl pH 8.5, 1 mM EDTA, 5 % glycerine, 1 mM β-mercaptoethanol, 10 μM dithiothreitol [DTT], 1 x protease inhibitor cOmplete^TM^. RIPA buffer: 50 mM Tris (pH 8), 150 mM NaCl, 1 mM EDTA, 1% NP-40, 0.5% sodium deoxycholate, 0.1% SDS. This buffer was stored at 4 °C before use. Prior to use, freshly prepared PMSF 100 mM stock (10 μL), Na_3_VO_4_ 200 mM stock (10 µL), DTT 1 M stock (2 μL) and 7x protease inhibitor (142.9 μL) were added to 835 µl RIPA buffer to get 1 mL working buffer.

### 5.7. Western Blot

Following the separation of proteins by sodium dodecyl sulphate polyacrylamide gel electrophoresis (SDS-PAGE) and transfer to nitrocellulose membranes, the following antibodies were used: Anti-β-actin (Abcam; Ab8227), anti-HSP90 (Cell Signaling Technology, Frankfurt, Germany; No. 4874), anti-p53 (Santa Cruz Biotechnology, Heidelberg, Germany; sc-126), anti-phospho-p53 (Ser15) (Cell Signaling Technology; No. 9284), anti-phospho-p53 (Ser46) (Becton Dickinson; No. 558245), anti-MGMT (Sigma-Aldrich; HPA032136). Proteins were detected using the Odyssey 9120 Infrared Imaging System (Li-Cor Biosciences, Lincoln, Nebraska, USA). The membrane was dried at room temperature in the dark and scanned with Odyssey. Image J was used for the quantification.

### 5.8. Autophagy Assay

The Cyto-ID kit (ENZO Life Sciences, Lörrach, Germany) was used for quantifying autophagy. Cells were seeded in 6 cm dishes, being careful that cells were confluent when harvesting. The supernatant from each sample was transferred to a 15 mL tube, cells were rinsed with PBS and trypsinized with 1 mL trypsin-EDTA and taken up in 1 mL fresh medium, which was transferred to a 15 mL tube for centrifugation (1000 rpm, 5 min). The pellet was resuspended in 2 mL PBS, washed again in PBS and resuspended in 0.25 mL DMEM with 5% FBS without phenol red and 0.25 mL diluted Cyto-ID solution was added to each sample. After resuspension, the samples were incubated 30 min at 37 °C in the dark. After centrifugation (1500 rpm, 5 min), the supernatant was discarded and the pellet was resuspended in 1 mL assay buffer. Samples were centrifuged (1500 rpm, 5 min) and resuspended in 0.5 mL assay buffer and transferred into FACS tubes. FACS Canto was employed for the measurement. The data were analysed using the BD FACSDiva software.

### 5.9. The γH2AX Foci Assay

For measuring DSBs, the γH2AX foci assay was employed. The cells were seeded in 6 cm dishes in plates containing sterile cover slips. When harvesting, the medium was discarded, the samples were washed twice with PBS and cells were fixed in ice cold methanol:aceton (7:3 stored at −20 °C), kept on 4 °C for exactly 9 min. The fixation solution was removed, samples were rinsed three times with PBS and 2 mL PBS was added to each dish to keep the cover slips wet. The cover slip was put into a 3 cm dish (the cells side up), blocked with blocking buffer (5% BSA in PBS with 0.3% Triton X-100) for 1 h, the other cover slip was stored at 4 °C as a backup. The blocking buffer was removed, 50 µl of γH2AX antibody (Cell Signaling; Cat. No. 9718s) (1:1.000 dilution of γH2AX in PBS with 0.3% Triton X-100) was added on the cover slip for overnight incubation at 4 °C. After 3 times PBS washing, 50 µl of the secondary antibody (Alexa Fluor^®^ 488, rabbit green, 1:500 of Alexa Fluor^®^ 488 in PBS with 0.3% Triton X-100) was added to the cells on the cover slip and incubated at room temperature in the dark for 2 h, followed by three times washing with PBS. The secondary antibody (Alexa Fluor^®^ 488) was from Life Technologies, Carlsbad, USA. DAPI-Vectashield (Vector Laboratories, Burlingame, CA, USA) and the solution (1.5 µl of 1 mg/mL DAPI was added in 1 mL Vectashield mounting medium, and vortexed thoroughly) was prepared freshly for staining. 20 µl of the DAPI-Vectashield solution was dropped on the center of one slide, the cover slip was put on the DAPI-Vectashield solution and sealed by nail oil. The slides were kept in the dark at room temperature for 10 min to dry. The γH2AX foci numbers were determined using the Metasystem finder version 3.1. Representative pictures of foci are shown in Appendix A.

### 5.10. Senescence Measurements with C_12_FDG Staining

Premature senescence was determined using C_12_FDG and flow cytometry quantification. In brief, C_12_FDG is a substrate of SA-β-galactosidase. Upon cleavage it produces a green fluorescence, which can be detected by FACS. Bafilomycin A1 is an inhibitor of vacuolar type H^+^-ATPase (V-ATPase). It blocks lysosomal acidification and also increases the pH of lysosomes [55]. Bafilomycin A1 (Sigma-Aldrich, Steinheim, Germany) was dissolved in DMSO at 0.1 mM stock solution and stored at −20 °C. The working concentration was 100 nM. C_12_FDG (Sigma-Aldrich, Steinheim, Germany) was dissolved in DMSO (20 mM stock solution) and stored at -20 °C. The stock solution was diluted with fresh medium to get a 2 mM working solution. Cells were seeded and treated 96 h before the assay was performed. They were incubated with 100 nM bafilomycin A1 for 1 h and thereafter with 33 µM C_12_FDG for 2 h. All the procedures after C_12_FDG incubation were operated avoiding light. The samples were rinsed with PBS three times, 30s each, harvested with trypsin-EDTA and resuspended in serum containing medium together with the cells in the supernatant. They were collected by centrifugation at 4 °C, 100–250 g, 5 min. The pellet was resuspended in 0.4–0.5 mL PBS (4° C) and cells (titer of about 1 × 10^6^/mL) were measured in a FACS Canto II flow cytometer.

### 5.11. Statistical Analysis

If not clarified specifically, data points show the means of at least three independent experiments and the standard deviation from mean as error bars. For comparison, two-way ANOVA was employed, the calculated *p*-values are displayed: *p*-value < 0.05 *, *p*-value < 0.005 **, *p*-value < 0.001 ***, *p*-value < 0.0001 ****. GraphPad Prism software was used for statistical analysis and graph plotting.

## Figures and Tables

**Figure 1 ijms-20-01562-f001:**
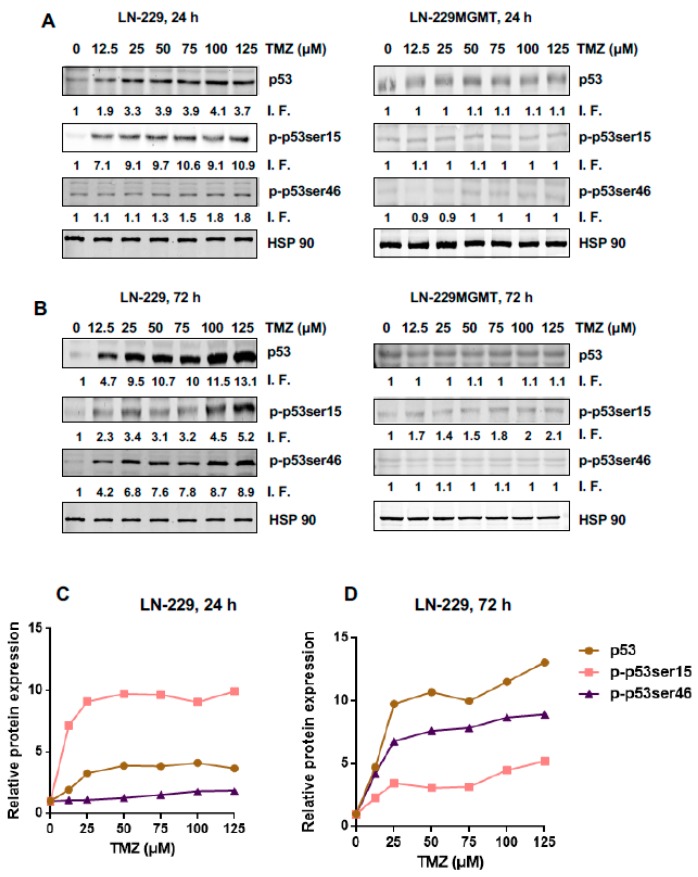
p53 expression and phosphorylation levels of p-p53ser15 and p-p53ser46 in LN-229 and LN-229MGMT cells treated with doses of TMZ up to 125 µM. (**A**) LN-229 and LN-229MGMT cells were exposed to doses of TMZ between 0 and 125 μM. 24 h later cells were harvested and p53 protein expression, phosphorylation levels of p53ser15 and p53ser46 (p-p53ser15 and p-p53ser46) were detected by Western blot. (**B**) LN-229 and LN-229MGMT cells were exposed to different doses of TMZ (0 μM-125 μM), 72 h later p53 protein expression, phosphorylation levels of p53ser15 and p53ser46 were detected by Western blot. HSP90 was used as the loading control. I.F. means induction factor. (**C**) Relative expression levels of total p53, p-p53ser15 and p-p53ser46 in LN-229 cells 24 h and (**D**) 72 h after addition of TMZ to the medium. The blots were quantified using the Odyssey image analysis system with ImageJ software and plotted on a liner scale.

**Figure 2 ijms-20-01562-f002:**
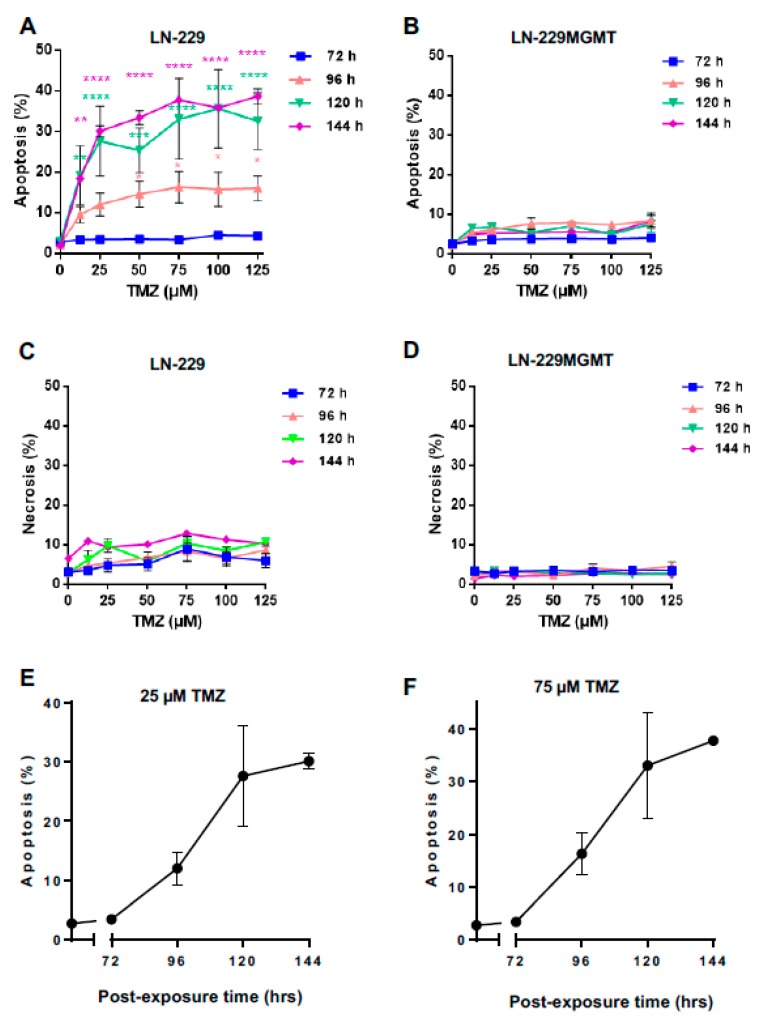
Apoptosis and necrosis levels in LN-229 and LN-229MGMT cells after TMZ treatment with doses up to 125 µM. (**A**,**B**) Apoptosis and (**C**,**D**) necrosis induced by TMZ in LN-229 and LN-229MGMT cells, detected by AV/PI double-staining and flow cytometry. Data were obtained 72, 96, 120 and 144 h after TMZ exposure. Data were analysed using BD FACSDiva and the Prism software. *p*-values of < 0.05 are marked as *, *p* < 0.01 as **, *p* < 0.001 as *** and *p* < 0.0001 as ****. (**E**,**F**) Apoptosis as a function of post-exposure time in LN-229 cells treated with 25 µM (panel **E**) and 75 µM TMZ (panel **F**).

**Figure 3 ijms-20-01562-f003:**
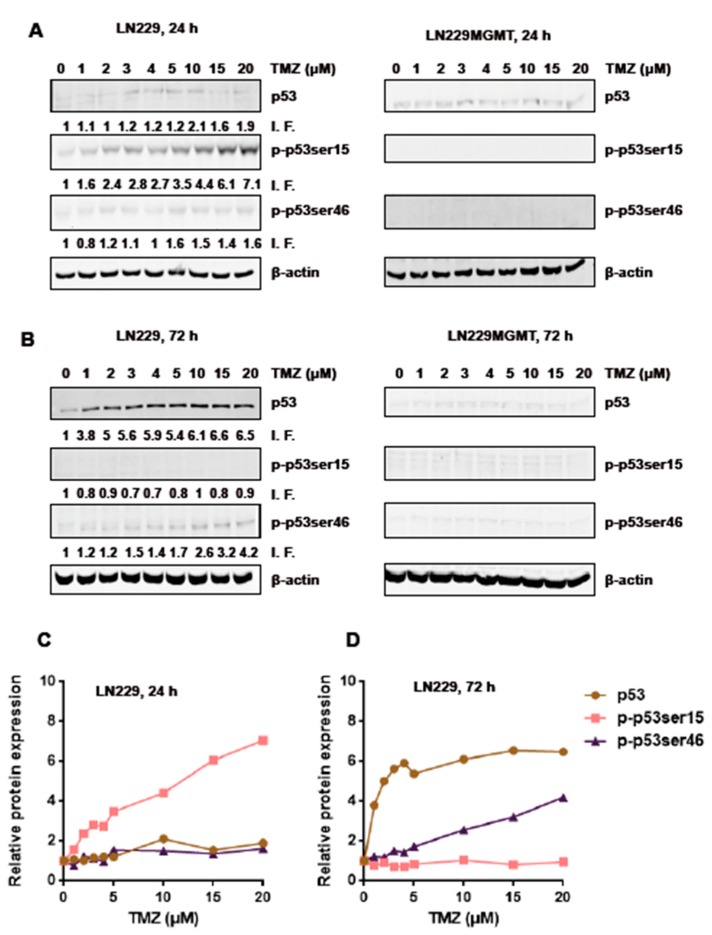
p53 expression and phosphorylation levels of p-p53ser15 and p-p53ser46 in LN-229 and LN-229MGMT cells treated with low doses of TMZ (up to 20 µM). (**A**) LN-229 and LN-229MGMT cells were exposed to different doses of TMZ and 24 h later cells were lysed onto the plates, protein extracts were obtained and total p53 protein and p-p53ser15 and p-p53ser46 were detected by Western blot analysis. (**B**) The same was performed 72 h after TMZ treatment. β-actin was used as loading control. I.F. means induction factor, which is related to the non-exposed control. (**C,D**) Relative expression levels of p53, p-p53ser15 and p-p53ser46 in LN-229 cells 24 and 72 h after TMZ treatment. Blots were quantified and analysed with ImageJ software. Data from representative experiments are shown.

**Figure 4 ijms-20-01562-f004:**
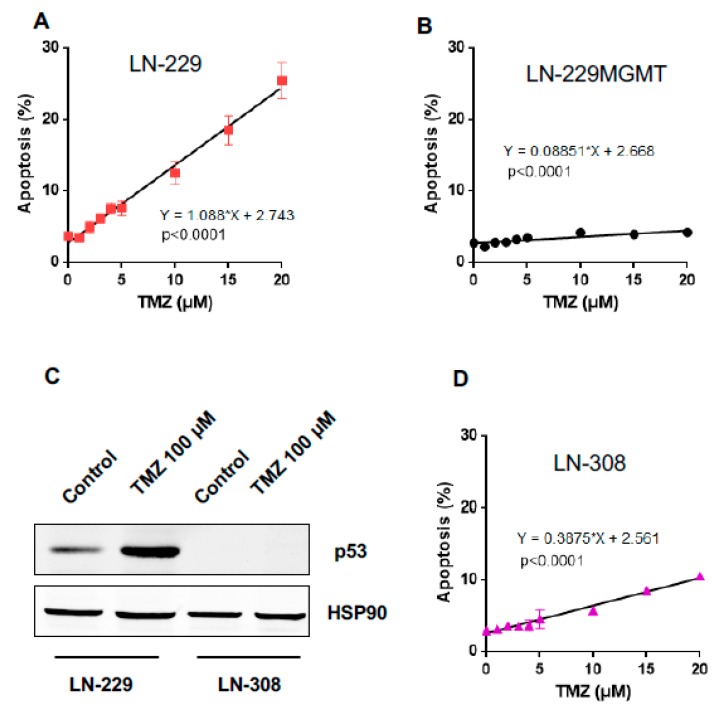
TMZ-induced apoptosis as a function of dose of TMZ dose in LN-229, LN-229MGMT and p53 lacking LN-308 cells. (**A**) Apoptosis as measured 120 h after TMZ exposure as a function of dose in LN-229 cells and (**B**) LN-229MGMT cells. Data are the mean of three independent experiments. (**C**) LN-229 and LN-308 cells were exposed to 100 μM TMZ, protein extracts were collected 72 h later and the p53 protein expression was detected by Western blot. HSP90 was used as loading control. (**D**) Apoptosis in LN-308 cells as a function of dose of TMZ measured 120 h after TMZ treatment. Linear regression analysis was done as described in Materials and Methods.

**Figure 5 ijms-20-01562-f005:**
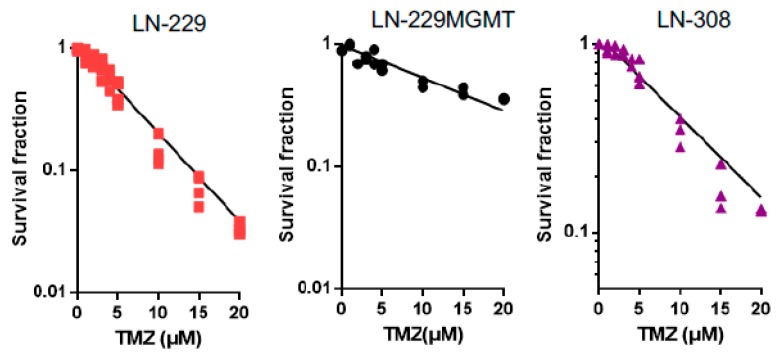
Survival (colony formation) as a function of TMZ dose in LN-229, LN-229MGMT and p53 lacking LN-308 cells. Colony survival assays with LN-229, LN-229MGMT and LN-308 cells exposed to different concentrations of TMZ. Regression lines are the best fit and drawn following computer analysis.

**Figure 6 ijms-20-01562-f006:**
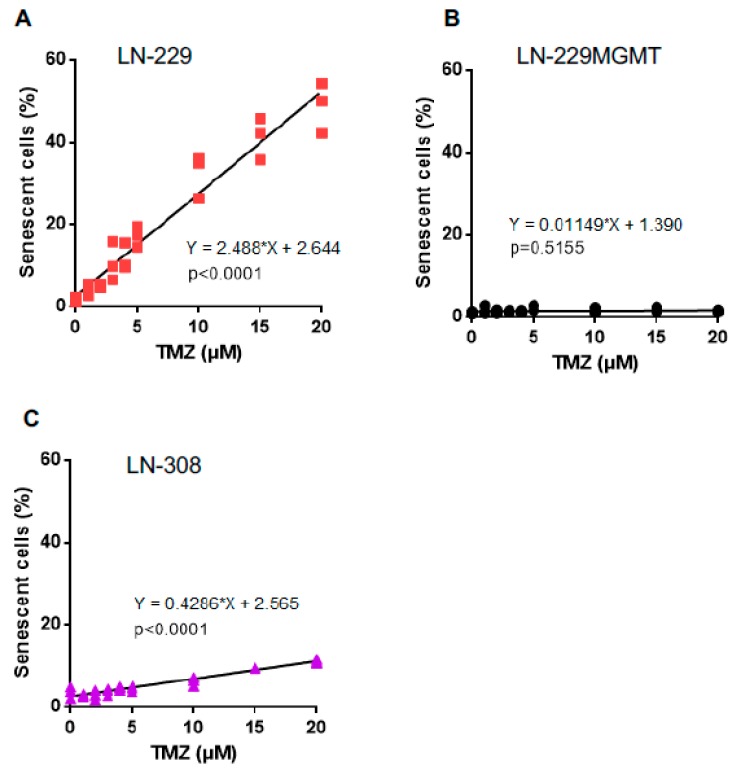
TMZ-induced senescence in LN-229, LN-229MGMT and LN-308 cells. (**A**) LN-229, (**B**) LN-229MGMT and (**C**) LN-308 cells were exposed to TMZ in the dose range of 0–20 μM. 96 h later cells were collected, stained with C_12_FDG for 15 min, and analysed by flow cytometry. Regression lines represent the best fit of data obtained.

**Figure 7 ijms-20-01562-f007:**
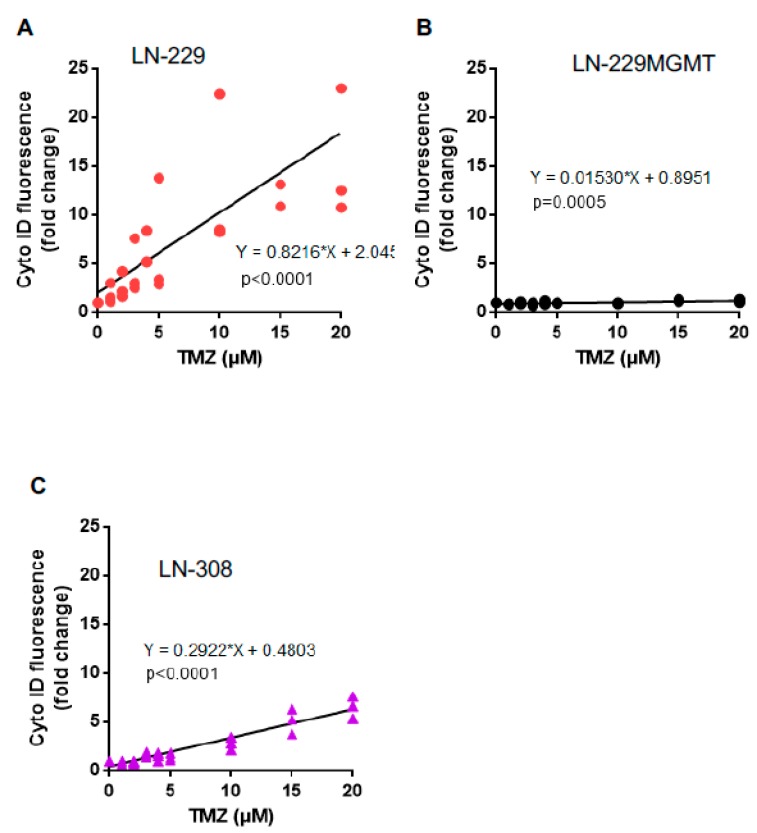
TMZ-induced autophagy in LN-229, LN-229MGMT and LN-308 cells. (**A**) LN-229, (**B**) LN-229MGMT and (**C**) LN-308 cells were exposed to TMZ in the dose range of 0-20 μM. 96 h later cells were collected and analysed. The results were shown by Cyto ID fluorescence fold change and compared with control samples. Control samples in the three cell lines were set to 1.

**Figure 8 ijms-20-01562-f008:**
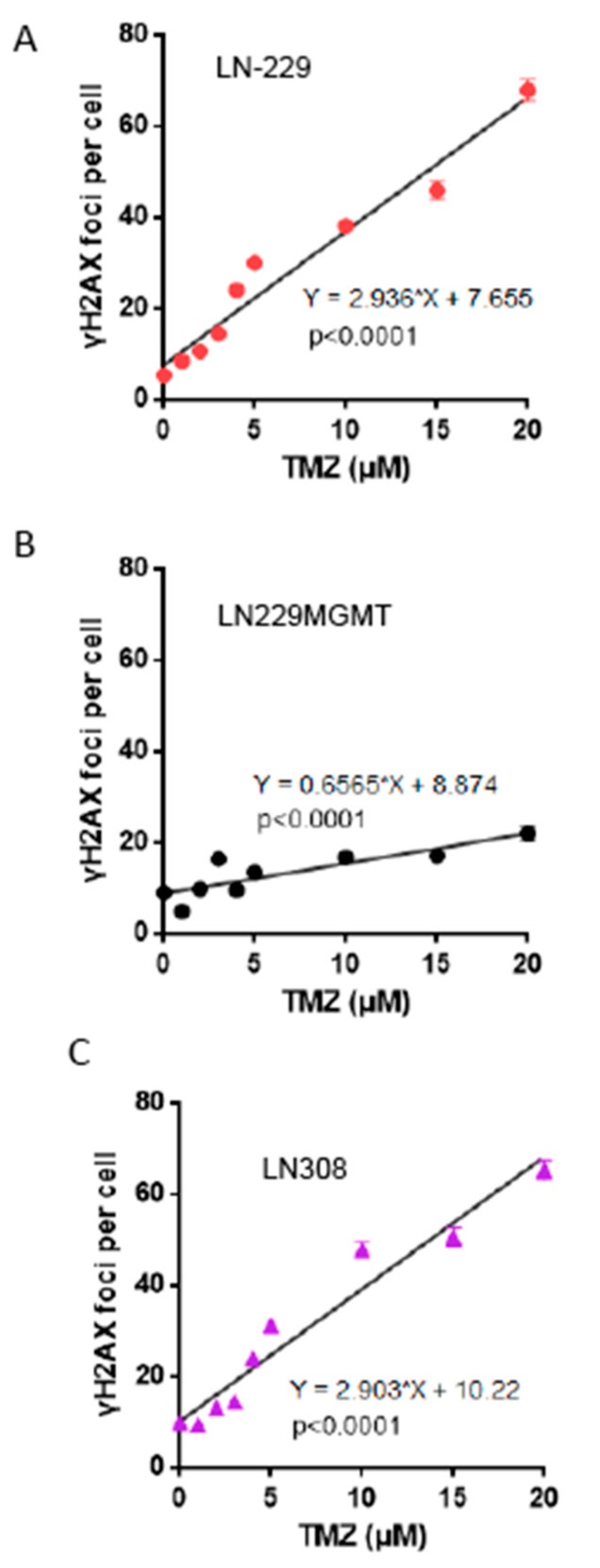
γH2AX foci as a function of dose of TMZ in LN-229, LN-229MGMT and LN-308 cells. Cells were exposed to TMZ in the dose range of 0-20 μM. 72 h later cells were fixed, stained for γH2AX and foci were quantified by the Metafer system in (**A**) LN-229, (**B**) LN-229MGMT and (**C**) LN-308 cells. Data are the mean of several experiments and were assessed by linear regression analysis.

**Figure 9 ijms-20-01562-f009:**
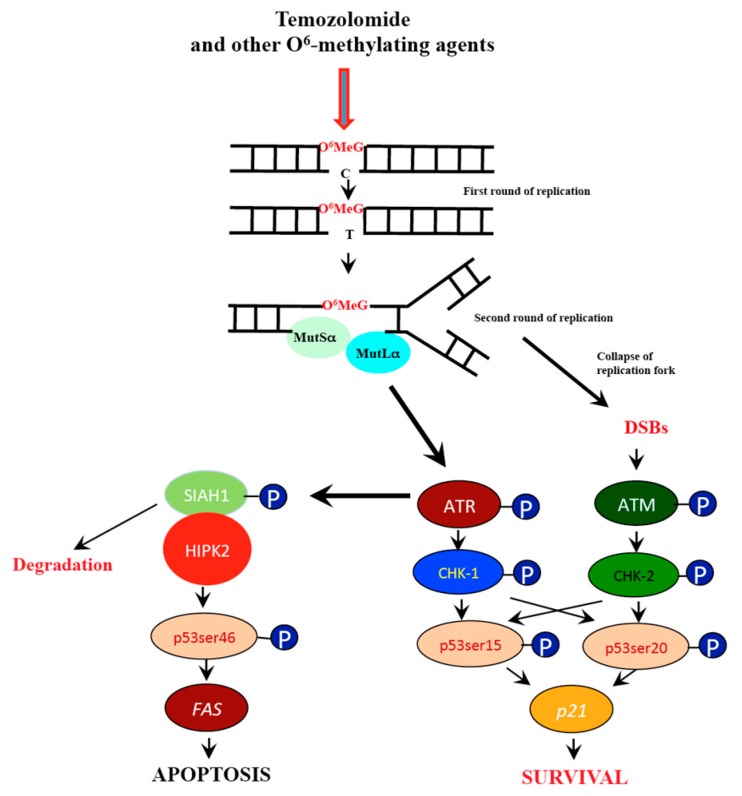
Pathways triggered by *O*^6^-methylguanine (a simplified schema). *O*^6^MeG is processed by MMR resulting in replication blockage and the formation of free DSBs. This gives rise to ATR and ATM and downstream p53 activation. The kinase HIPK2 becomes activated following SIAH1 phosphorylation, which in turn activates p53ser46 that targets the promoter of several pro-apoptotic genes. Although p-p53ser15 can transcriptionally activate DNA repair genes such as *DDB2*, which is involved in nucleotide excision repair, there is no evidence yet that p53 upregulates DNA repair genes (as an immediate-early response) that are of importance for temozolomide resistance.

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
