# Peer review of "Are There Thresholds in Glioblastoma Cell Death Responses Triggered by Temozolomide?"

_ijms, 2019, doi:10.3390/ijms20071562_

Reviewer 1 Report

The article “Are there thresholds in glioblastoma cell death responses triggered by temozolomide?” by Y. He and B. Kaina investigates the response of several cell lines to TMZ. The article is well written and organized and, in my opinion, only minor formatting and editing errors must be addressed before publication (consistency in the bibliography, chemical names of compounds, ml is mL, spaces before °C, rpm must be converted to rcf, etc).

Also, error bars should be added to all graphs (where appropriate), i.e. Fig. 1 c-d, Fig. 3 c-d, etc.

Other more specific points:

Line 79: “und” instead of “and”.

Line 87: “manuscript submitted” should be updated with data of the accepted manuscript (Cancer Res?). The same applies to lines 425 and 440 (He et al. in press 2019).

Line 117: include the reference in the References section.

Lines 397-400: please revise the sentence.

Titles in the Materials and Methods section must be formatted properly.

Author Response

The article “Are there thresholds in glioblastoma cell death responses triggered by temozolomide?” by Y. He and B. Kaina investigates the response of several cell lines to TMZ. The article is well written and organized and, in my opinion, only minor formatting and editing errors must be addressed before publication (consistency in the bibliography, chemical names of compounds, ml is mL, spaces before °C, rpm must be converted to rcf, etc).

Thank you. 

Also, error bars should be added to all graphs (where appropriate), i.e. Fig. 1 c-d, Fig. 3 c-d, etc.

Figure 1C,D and Fig. 3C,D are quantifications of blots shown in the same figure. These are representative examples. Western blots are complex experiments with a lot of variables, for which it is not usual to operate with mean values resulting from different experiments. All experiments displayed upregulation. In the legend of Fig. 1 and Fig. 3, we emphasize that representative blots and their quantification are shown.

Other more specific points:

Line 79: “und” instead of “and”.

Corrected

Line 87: “manuscript submitted” should be updated with data of the accepted manuscript (Cancer Res?). The same applies to lines 425 and 440 (He et al. in press 2019).

Meanwhile the paper is in press (Mol. Cancer Res.) and online available. The reference is included now.

Line 117: include the reference in the References section.

This reference is deleted because it was unnecessary.

 Lines 397-400: please revise the sentence.

Has been done.

Titles in the Materials and Methods section must be formatted properly.

Has been done.

 Reviewer 2 Report

The manuscript is well-written, easy-readable and accurately demonstrates results. However, this study is performed exclusively in vitro with only one major cell line (LN-229 cells and its derivative LN-229MGMT) and results should be interpreted with grate caution since it is well-know that, for example, the long-term TMZ treatment induces unpredictable genetic and phenotypic changes across treated cell lines in vitro and even clone specific-responses (see doi: 10.1186/s12935-016-0311-8 or PMID:27158244). Thereby, cell-type dependent responses may be relevant for this type of study.

Major points

1) Lines 436-437, “In accordance with this is the finding that apoptosis (Fig. 4) and reproductive death (Fig. 5) of LN-229 cells does not display a threshold; is a linear function of dose of temozolomide”. It is true for the TMZ concentrations ranging 0-20 µM. However, in figure 2, the authors clearly demonstrated a threshold for TMZ concentration in apoptosis induction. After 25 µM dose, TMZ had no effect on apoptosis in LN-229 cells even after 120-144 hrs. The authors should explain clearly, what they mean under a term “threshold” in their study.

2) Lines 462-464, the authors state “Although we are aware that this study needs extension to other cell lines and tumor models, LN-229 provides an example that the above paradigm is not generally true… Further studies are required with a larger set of well-defined cell lines and with tumors in situ prior to and after therapy.” If we wish to get a picture as whole, it would be appropriate to extend this study to a panel of cell lines before to make any significant conclusions and, moreover, recommend clinical implications. The authors should include several additional cell lines in their study before the manuscript being considered for publication.

3) Lines 477-499, regarding the clinical implications, this study has no obvious clinical implications. The patients are treated according to the Stupp protocol (concurrent radiotherapy (RT, 60 Gy delivered in 30 fractions, five times a week for 6 weeks, fractions of 2 Gy each) and daily oral TMZ (75 mg/m2 per day, given 7 days per week), and then, after 4 treatment-free weeks, adjuvant TMZ up to 6 cycles (150–200 mg/m2 per day, for five consecutive days, 28-day cycle). The authors should take into account that the peak TMZ concentrations in plasma, cerebrospinal fluid (CSF), and the brain extracellular fluid of glioma patients ranged from 3 to 15 µg/mL (15-77 µM), 0.16 to 1.93 μg/ml (0.8-9.9 µM), and 0.3 to 0.9 µg/mL (1,5-4,6 µM), respectively (Patel et al., 2003; Portnow et al., 2009). Up to 5 µM TMZ concentration increases patient survival very moderately. Based on the results of the authors, 5 µM TMZ concentration had a marginal effect on apoptosis, autophagy and senescence induction in LN-229 cells. It would be interesting to carry out the similar study on a panel of glioma cell lines using metromomic TMZ (drug application at low and frequent doses) and a non-standard low dose radiotherapy fractionation (≤0.1 Gy).

 Minor points

1) In the Western blot section, the authors must mention the catalogue numbers of the antibodies (“Anti-b-actin, anti-HSP90 and anti-p53 are all from Santa Cruz Biotechnology (Heidelberg, Germany), anti-phospho-p53 (Ser15) and anti-phospho-p53 (Ser46) are from Cell Signalling Technology (Frankfurt, Germany), anti-MGMT from Chemicon International Inc.”)

2) Lines 397-399, a fragment “account for approximately 60 to 70% of high-grade malignant glioma” in the sentence is duplicated. Please, correct it. (“Despite this well-known relationships, the prognosis of glioblastomas, which account for approximately 60 to 70% of high-grade malignant glioma, accounts for approximately 60 to 70% of high-grade malignant gliomas is bleak, as the median length of survival is only 12 to 15 mounts [24]”). The reference 24 in this sentence states “the median length of survival is only 12 to 15 mounts” in glioblastoma patients. However, the recent phase III clinical trials showed that the median overall survival for adult patients with newly diagnosed glioblastoma in the control arms has reached up to 18-20 month. These data might explain a failure of positive non-controlled phase II trials to predict positive phase III trials and should result in revision of the landmark Stupp trial as a historical control for median overall survival in non-controlled trials (please, for discussion see DOI: 10.3390/cancers10120492 or PMID: 30563098).

3) Line 464, “Regarding DNA repair, is known…” should be probably “it is known”

4) Lines 466-467, “However, these genes resp. the corresponding proteins are not involved in the repair of temozolomide-induced DNA methylation damage”. What does “these genes resp.” mean? Please, correct.

Author Response

The manuscript is well-written, easy-readable and accurately demonstrates results. However, this study is performed exclusively in vitro with only one major cell line (LN-229 cells and its derivative LN-229MGMT) and results should be interpreted with grate caution since it is well-know that, for example, the long-term TMZ treatment induces unpredictable genetic and phenotypic changes across treated cell lines in vitroand even clone specific-responses (see doi: 10.1186/s12935-016-0311-8 or PMID:27158244). Thereby, cell-type dependent responses may be relevant for this type of study.

We partly agree. It is clear that the study has a limitation, i.e. most of the experiments were performed with the isogenic cell pair LN-229 and LN-229MGMT (p53wt) and the line LN-308 (p53mt). We strongly emphasize this limitation and notify that it needs extension with many more lines and also stem cells and sphere cultures in order to arrive at a general conclusion. Nevertheless, it is worth to publish the data because it provides an example that cell death and other end points like senescence and autophagy increase with dose without a no-effect threshold. This observation is remarkable because the general believe is that in the low dose range cells are protected because of DNA repair and start to die only at high doses. This concept is obviously not (generally) true. 

I do agree with the notion that TMZ treatment causes genetic and phenotypic changes. That is to be expected since TMZ is a strong point mutagen. However, to my knowledge LN-229 and LN-308 were obtained from pretreatment GBM. The cells have not been treated before with TMZ. This, however, does not exclude cell type specific differences. 

 Major points

1)     Lines 436-437, “In accordance with this is the finding that apoptosis (Fig. 4) and reproductive death (Fig. 5) of LN-229 cells does not display a threshold; is a linear function of dose of temozolomide”. It is true for the TMZ concentrations ranging 0-20 µM. However, in figure 2, the authors clearly demonstrated a threshold for TMZ concentration in apoptosis induction. After 25 µM dose, TMZ had no effect on apoptosis in LN-229 cells even after 120-144 hrs. The authors should explain clearly, what they mean under a term “threshold” in their study.

This notion rests likely on a misunderstanding. In Fig. 2, the time course of apoptosis is shown for cells treated with different doses. After post-incubation times of > 96 h cells undergo apoptosis with all doses used for treatment. 

The more relevant experiment is shown in Fig. 4A and 4D, demonstrating the dose-response for apoptosis over a broad dose range. The regression line represents the optimal fit, which clearly shows lack of a no-effect threshold and linearity in the response.

 2)     Lines 462-464, the authors state “Although we are aware that this study needs extension to other cell lines and tumor models, LN-229 provides an example that the above paradigm is not generally true… Further studies are required with a larger set of well-defined cell lines and with tumors in situ prior to and after therapy.” If we wish to get a picture as whole, it would be appropriate to extend this study to a panel of cell lines before to make any significant conclusions and, moreover, recommend clinical implications. The authors should include several additional cell lines in their study before the manuscript being considered for publication.

See my comment above. I should emphasize that I do not arrive at general conclusions and stress the limitation of the study. The data was discussed with great caution, always in view the limitations.  -  Quantitative western blots are not trivial, and the markers we were using are difficult to determine (notably p53ser46). Therefore, extending the work to many other cell lines including stem cells needs 1-2 year of work. I hope that this study will initiate further studies in order to substantiate the data, deducing more general conclusions.

Regarding clinical implications: I tempered this section. Of note, we do not give any recommendation for treatment etc. It is just stated that, if a no-effect threshold does not exist, even low doses will have an effect. And, as dicussed, this is a new view at an old issue.

 3) Lines 477-499, regarding the clinical implications, this study has no obvious clinical implications. The patients are treated according to the Stupp protocol (concurrent radiotherapy (RT, 60 Gy delivered in 30 fractions, five times a week for 6 weeks, fractions of 2 Gy each) and daily oral TMZ (75 mg/m2per day, given 7 days per week), and then, after 4 treatment-free weeks, adjuvant TMZ up to 6 cycles (150–200 mg/m2per day, for five consecutive days, 28-day cycle). The authors should take into account that the peak TMZ concentrations in plasma, cerebrospinal fluid (CSF), and the brain extracellular fluid of glioma patients ranged from 3 to 15 µg/mL (15-77 µM), 0.16 to 1.93 μg/ml (0.8-9.9 µM), and 0.3 to 0.9 µg/mL (1,5-4,6 µM), respectively (Patel et al., 2003; Portnow et al., 2009). Up to 5 µM TMZ concentration increases patient survival very moderately. Based on the results of the authors, 5 µM TMZ concentration had a marginal effect on apoptosis, autophagy and senescence induction in LN-229 cells. It would be interesting to carry out the similar study on a panel of glioma cell lines using metromomic TMZ (drug application at low and frequent doses) and a non-standard low dose radiotherapy fractionation (≤0.1 Gy). 

Thank you for this valuable information. I added a short paragraph discussing this point and adding the reference Portnow (Patel is a nice work, but with primates) and others.

 Minor points

1)     In the Western blot section, the authors must mention the catalogue numbers of the antibodies (“Anti-b-actin, anti-HSP90 and anti-p53 are all from Santa Cruz Biotechnology (Heidelberg, Germany), anti-phospho-p53 (Ser15) and anti-phospho-p53 (Ser46) are from Cell Signalling Technology (Frankfurt, Germany), anti-MGMT from Chemicon International Inc.”)

This has been done.

 2)     Lines 397-399, a fragment “account for approximately 60 to 70% of high-grade malignant glioma” in the sentence is duplicated. Please, correct it. (“Despite this well-known relationships, the prognosis of glioblastomas, which account for approximately 60 to 70% of high-grade malignant glioma, accounts for approximately 60 to 70% of high-grade malignant gliomas is bleak, as the median length of survival is only 12 to 15 mounts [24]”). The reference 24 in this sentence states “the median length of survival is only 12 to 15 mounts” in glioblastoma patients. However, the recent phase III clinical trials showed that the median overall survival for adult patients with newly diagnosed glioblastoma in the control arms has reached up to 18-20 month. These data might explain a failure of positive non-controlled phase II trials to predict positive phase III trials and should result in revision of the landmark Stupp trial as a historical control for median overall survival in non-controlled trials (please, for discussion see DOI: 10.3390/cancers10120492 or PMID: 30563098).

Thank you again for this valuable information. I discussed this point more thoroughly and added the corresponding references.

 3)     Line 464, “Regarding DNA repair, is known…” should be probably “it is known”

Thank you. Corrected.

 4)     Lines 466-467, “However, these genes resp. the corresponding proteins are not involved in the repair of temozolomide-induced DNA methylation damage”. What does “these genes resp.” mean? Please, correct.

Thank you. Corrected.

Round  2

Reviewer 2 Report

The authors have responded to all queries and introduced some modifications in their manuscript. However, the principal issue has not been resolved.

In the manuscript, the authors “are aimed at proving evidence for or disproving the widely accepted general paradigm that low DNA damage levels induce survival functions, whereas high DNA damage levels activate cellular death pathways [1]… The concept implicates that there are threshold doses for cell death, i.e., low doses do not elicit activation of apoptosis pathways while high doses do”. In conclusion, the authors state that “LN-229 provides an example that the above paradigm is not generally true”. However, based on the provided data by the authors, it seems for me, the opposite conclusion is true! The tumor cell lines are highly genetically and phenotypically heterogeneous. Within a cell line, each individual tumor cell, based on single-cell analysis, is unique. In each individual cell, the different levels of expression and activity of pro-apoptotic/anti-apoptotic and DNA repair proteins are expected. Each individual cell has its own threshold dose for cell death. Increasing TMZ concentration (e.g., 1µM, 2µM, 3µM, 4µM, 5µM, 10µM, 15µM, and 20 µM as the authors have used) leads to the incremental proportions of apoptotic LN-229 cells, since a threshold dose for cell death is reached in an increasing number of cells within a heterogeneous population. Single-cell analysis such as colony formation assay clearly demonstrated that different concentrations of TMZ eradicated different amounts of colony-forming units, supporting evidence that a different threshold do exist for each individual cell/clone. Second, the TMZ concentrations ranging 0-125 µM did not induce apoptosis in MGMT-overexpressing LN-229MGMT isogenic cells in comparison to parental LN-229 cells, again supporting evidence that a threshold exists. MGMT expression modifies a threshold, significantly increasing it. It might require 500-1000 µM TMZ to induce apoptosis in MGMT-overexpressing LN-229MGMT cells at the simillar level to that observed in LN-229.

The fact that TMZ at low doses has an inhibitory/suppressing/death-promoting effect against MGMT-negative cells is not novel. In this sense, your study adds little.

The problem of the study design is that in almost all assays, thousands/millions of cells are analyzed, and the “average results" within heterogeneous population are acquired. However, only single-cell/clone analysis could provide firm evidence whether there is a threshold in apoptosis induction.

See additionally 1) doi 10.3390/cancers11020190, 2) doi 10.1021/acs.nanolett.6b00902

3) doi 10.1073/pnas.1320611111  4) doi 10.1186/1479-5876-12-184

If I am wrong in my discussion, please, provide clear and detailed explanation with arguments against.

Furthermore, in figure 2A, the authors clearly demonstrated that after 25 µM dose, TMZ had no effect on an increase of apoptosis in LN-229 cells. Why does 25µM TMZ induce the same level of apoptosis (at least, very simillar) as 125 µM? This has not been discussed in the manuscript.

Minor corrections

1) Lines 405-408. These sentences are not logically linked. Please, correct.“It is effective in inducing cell death if the tumor lacks MGMT or expresses it at low level, i.e. < 30 fmol/mg protein [23]. These tumors are defined as “methylated” since MGMT promoter CpG methylation correlates with silencing of the gene [24] and, consequently, deficient or low MGMT protein expression and enzyme activity [25].”

2) Lines 509-510. Please, correct “is be”. “Nevertheless, the data may be taken to indicate that even a low dose of temozolomide might is be able to elicit a therapeutic cytotoxic response in p53 wild-type and MGMT lacking tumors”.

Author Response

The repeated reply of reviewer # 2 is an indication of the importance of the topic that we addressed in this manuscript. It also indicates to me that there are misunderstandings leading to the objections expressed by the reviewer.

a) The major concern in this second reply pertains the question of thresholds. The term “threshold” is defined in Toxicology as a dose below which no adverse effects can be observed, or more precisely, below which there is a low probability for a given effect that is under study. This results in "No observed effect level" (NOEL),  “No observed adverse effect level” (NOAEL) or “No observed genotoxic effect level” (NOGEL), i.e. thresholds are always related to a given endpoint. For assessing whether a threshold for a given endpoint (e.g. death of mice following treatment with increasing doses of a toxin or genotoxicant or inflammation, cancer formation, weight loss, etc.) dose-response curves have to be established. From these, a threshold dose for a given endpoint can be calculated. If the dose-response is linear, a threshold for a given effect cannot be deduced, i.e. there is no threshold. If the optimal fit of a dose-response curve is not linear, a threshold can be calculated below which the effect is not observable (NOEL).

According to this, from the toxicological point of view, a threshold cannot be defined or determined for a single individual or a single cell. Thresholds are defined for exposed populations (plants, animals, man, cells) as they indicate the probability for the appearance of a given effect with increasing dose.

Having said this, I disagree with the reviewer that “each individual cell has its own threshold dose for cell death”. It is true that on single cell level, the dose that kills a particular cell is to some extent variable, which makes the variability in the population. There might be more resistant and more sensitive cells in the population and the variability may be large or low (stochastic processes do also play a role), but this not necessarily results in a threshold (as defined by toxicologists). It rather has an impact on the shape of the dose-response curve. The shape can also be hockey-stick-like, which is an indication of the presence of resistant and sensitive subpopulations. A threshold, however, results from complete defending or tolerating a poison, up to a dose above toxicity (or other effects) is emerging. Usually, genuine poisons (e.g. cyanide) exert a threshold (at a subtoxic dose all individuals survive; so we can be happy not being poisoned if we eat 5 peach pits), while carcinogens are considered to exhibit no threshold (although this paradigm is currently under debate). 

Taken together, the variability in a given (cell) population, which can be measured on single cell level (e.g. colony formation but also apoptosis-flow as single cells are counted and other methds), is not identical to a threshold dose (for a given endpoint) that pertains to the whole cell population.

Regarding the dose-response curves: Our work was not devoted to measure the variability within the population of, for e.g., LN-229 cells. It is clear that there is some variability, although we are impressed how stable the lines we are working with are, otherwise reproducible studies in many labs worldwide wouldn't be achievable. But despite intercellular variability, thresholds can be defined for a given population. Our dose-response curves for the endpoints we have studied are linear (this is the best fit) and, therefore, a NOEL cannot be derived. This is the case for dose-response curves in the colony forming assays, apoptosis and the other endpoints we have measured.

b) In this second argument, the reviewer refers to LN-229MGMT cells. These cells, as a population, display a no-effect threshold, and I think we made very clear that this is the case (see Discussion): "A reasonable candidate causing a threshold is MGMT. Thus, from work with bacteria to man it became clear that MGMT mediated DNA repair gives rise to a mutagenic and toxic threshold [1-3]." 

In our previous papers, we demonstrated a key role of MGMT in setting up a threshold, i.e. no effect at low-dose levels. We showed this for the endpoint colon carcinogenesis in MGMT ko mice [4], skin cancer in the transgenic mouse model [5]and discussed it repeatedly, also in the context of nonlinear dose-responses [16]. 

The main reason for including LN-229MGMT in this study was to assess whether MGMT has an effect on p53 stabilization and p-p53ser15 and p-p53ser46 activation, which are considered key prosurvival and proapoptosis marker, respectively. The comparison of LN-229 and LN-229MGMT revealed that the p-p53ser15 and ser46 pathway is triggered by O6MeG and there is obviously no threshold in LN-229, neither for this p53 pathway nor for γH2AX and apoptosis (we added a schematic presentation of the pathway in Fig. 9 to make this more transparent). These are important findings and we obtained them only by a comparison of MGMT- versus MGMT+ cells. Therefore, LN-229MGMT is essential for any study that addresses the pathway activated by O6MeG, including senescence and autophagy. I'm not aware of any dose-response studies with these endpoints included and, thus, I can only reject the statement that the data are not novel.

c) Regarding the dose-response shown in Fig. 2. We show this data because, in many studies, authors were using 100 μM temozolomide or even more. The dose-response curve in this dose range (0-100 µM) shows that LN-229 are quite sensitive to temozolomide, reaching a saturation level at doses above 50 μM. What is the reason for the plateau shaped curves? DNA damage (also damage induced by temozolomide) induces not only apoptosis, but also necrosis, autophagy and senescence [7]. It is, therefore, not possible, to achieve 100% yield of apoptosis. The “saturation” level depends on the cell type and the genotoxin used. About 40% of apoptosis is, for LN-229 treated with TMZ, the maximum level. We added a sentence explaining the “saturation level”.

 Minor corrections: Thank you. This has been done.

 References

[1] A.D. Thomas, J. Fahrer, G.E. Johnson, B. Kaina, Theoretical considerations for thresholds in chemical carcinogenesis, Mutat Res Rev Mutat Res, 765 (2015) 56-67.

[2] B. Kaina, K. Ochs, S. Grosch, G. Fritz, J. Lips, M. Tomicic, T. Dunkern, M. Christmann, BER, MGMT, and MMR in defense against alkylation-induced genotoxicity and apoptosis, Prog Nucleic Acid Res Mol Biol, 68 (2001) 41-54.

[3] B. Kaina, M. Christmann, S. Naumann, W.P. Roos, MGMT: key node in the battle against genotoxicity, carcinogenicity and apoptosis induced by alkylating agents, DNA Repair (Amst), 6 (2007) 1079-1099.

[4] J. Fahrer, J. Frisch, G. Nagel, A. Kraus, B. Dorsam, A.D. Thomas, S. Reissig, A. Waisman, B. Kaina, DNA repair by MGMT, but not AAG, causes a threshold in alkylation-induced colorectal carcinogenesis, Carcinogenesis, 36 (2015) 1235-1244.

[5] K. Becker, A.D. Thomas, B. Kaina, Does increase in DNA repair allow "tolerance-to-insult" in chemical carcinogenesis? Skin tumor experiments with MGMT-overexpressing mice, Environ Mol Mutagen, 55 (2014) 145-150.

[6] A.D. Thomas, G.J. Jenkins, B. Kaina, O.G. Bodger, K.H. Tomaszowski, P.D. Lewis, S.H. Doak, G.E. Johnson, Influence of DNA repair on nonlinear dose-responses for mutation, Toxicol Sci, 132 (2013) 87-95.

[7] A.V. Knizhnik, W.P. Roos, T. Nikolova, S. Quiros, K.H. Tomaszowski, M. Christmann, B. Kaina, Survival and death strategies in glioma cells: autophagy, senescence and apoptosis triggered by a single type of temozolomide-induced DNA damage, PLoS One, 8 (2013) e55665.
